# [Re]Domain Generalization with MixStyle

1  ## Reproducibility Summary

2  **Scope of Reproducibility**

3  This work aims to confirm the effectiveness of MixStyle[8], which suggests mixing instance-level feature statistics of
4  training samples across source domains in order to generalize the model to unseen domain in domain generalization,
5  by achieving competing results on classification task. Furthermore, we extend the method to MixAll to improve the
6  stability, which combine feature statistics across all source domains at each step.

7  **Methodology**

8  All experiments were conducted on PACS[5] using ResNet-18[3] or ResNet-50. A single V100 or K80 is used for
9  experiments with an average training time of 2 hours, nearly 80 hours in total. We have reproduced MixStyle mainly
10  based on the paper using PyTorch and refer to the official implementation(Dassl.pytorch) for details and replicated
11  similar results shown in the last row of Table 1 in [8]. Ablation studies were focused on answering the question where
12  to apply MixStyle. We have implemented and supplemented most of the experiments on classification task.

13  **Results**

14  Compared with the average accuracy reported in in MixStyle, our implementation achieves similar accuracy but differs
15  a lot when switching model selection strategies. Therefore it is supported that MixStyle has a comparable results over
16  other methods like L2A[9] , considering MixStyle greatly saves computational resources and time. And the results of
17  our additional experiments demonstrate the method's limitations that the performance is sensitive to hyper-parameters
18  of $\alpha$ and the place to apply MixStyle. Furthermore, our MixAll method achieves more stable results under different
19  model selection strategies.

20  **What was easy**

21  It is generally easy to re-implement MixStyle given the idea and pseudo-code in the paper, with key idea and motivation
22  stated clearly and completely. And we have borrowed the data pre-processing code segment from Dassl.pytorch, which
23  save us much time.

24  **What was difficult**

25  The official implementation assembled in Dassl.pytorch is complicated for us students, thus spending a lot of time
26  to go through the library. Since experimenting with the official hyper-parameters would cost a lot of time, which
27  contradicting the author saying that this method saves time, we tried to use adam as optimizer with 30 epochs and come
28  out similar results. Besides, it is difficult for us to follow three totally different tasks including category classification,
29  instance retrieval and reinforcement learning, for the other two tasks requires four GPUs, so we could only experiment
30  on category classification with knowledge and computational resources jointly limited.

31  **Communication with original authors**

32  No communication with the original authors was required to reproduce their work.

# 1  Introduction

Convolutional neural networks(CNNs) have boosted a great success in computer vision over the past few years. However, the ability of CNN is largely limited when a trained model meets out-of distribution test data, which is commonly met in real world. To strengthen the generalization ability of CNNs, diverse source data from multiple relevant heterogeneous domains is collected so that CNN model is allowed to learn more domain-invariant features, and hence generalize to unseen out-of-distribution data, which is named unseen target domain. This problem is largely studied under domain generalization(DG).

Among recent DG methods, a widely used assumption is that images are generated from style and semantic information or disentangled into style features and semantic features, whose semantic information is domain-agnostic while the style information is domain-specific. As a result, the more semantic information a model learns, the stronger generalization ability the model will have. For instance, data augmentation is proven to be useful for enhancing the generalization ability of the model, such as learning to generate more novel domains using given source domains[9] or reconstructing images with mixed amplitude information and original phase information by a fourier-based framework[7]. However, model-based data augmentation requires more computational resources and training time, and fourier-based method is proposed in an innovative view thus hard to follow. The state-of-the-art method STEAM[1] belongs to disentanglement method, retaining style-invariant information and separating style from semantic information through a contrastive learning framework.

Observing that the first three outputs of residual block contain domain-related information as Figure1 while the last residual block encodes label-related information as Figure2, MixStyle[8] is proposed to mix the domain-related features directly in the similar way borrowed from AdaIN[4]. It is a plugin module playing the same role as dropout. The place to apply MixStyle is after the residual block while dropout is commonly applied after linear layers. Under domain generalization, however, one of the most important issue is where to apply Mixstyle.

In original paper, MixStyle appears to work in three totally different tasks including category classification, instance retrieval and reinforcement learning. For the sake of research interest and knowledge limitation, we reproduce the classification task on PACS dataset with sufficient experiments to exhibit how MixStyle works in CNNs and improves performance of classification.

Furthermore, we are not only re-implementing MixStyle, but also extend MixStyle to MixAll. According to the assumption that mixing style information leads to better generalization ability, we are motivated to explore whether mixing all statistic features of the provided source domains would perform better.

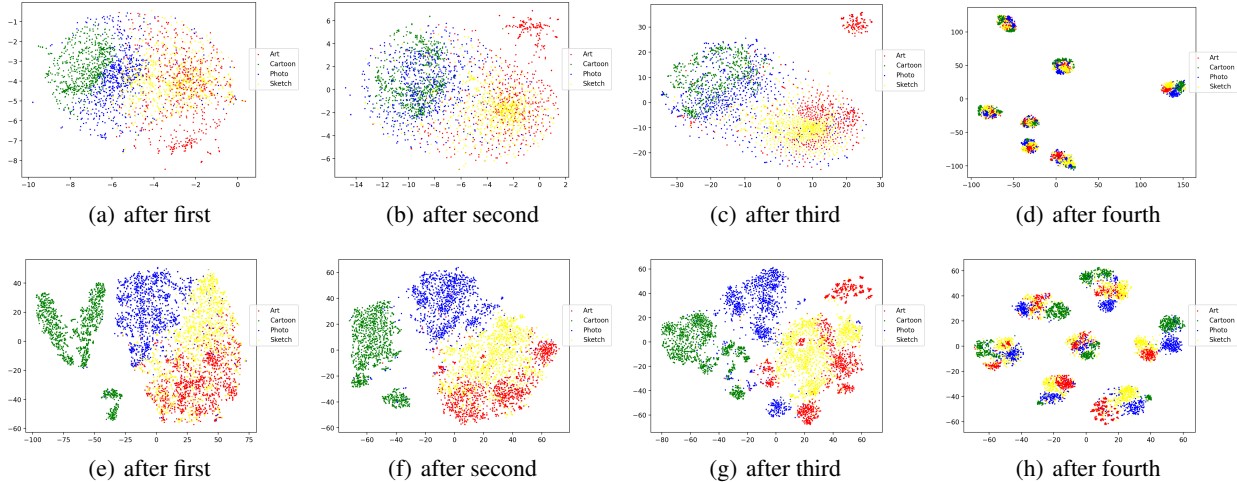

Figure 1: 2-D visualization of flattened feature maps(top) and the corresponding style statistics(bottom) with respect to domain labels. res1-4 denote the four residual blocks in order in a ResNet architecture. We observe that res1 to res4 all contain domain-related information, but res4 further encodes category information. Although domain-related information still remains after res4, label-related information lead the feature representation

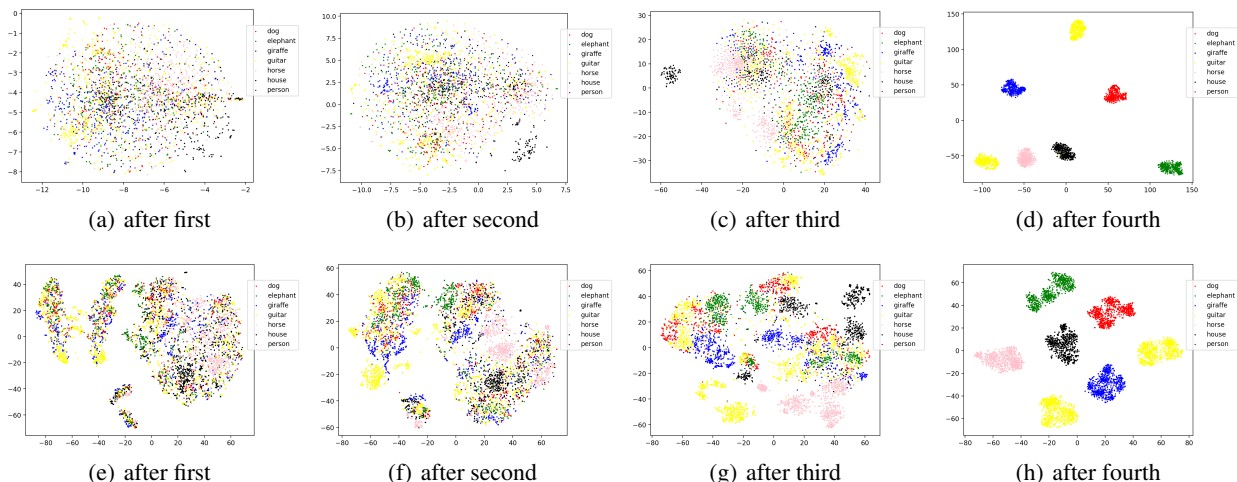

Figure 2: 2-D visualization of flattened feature maps(top) and the corresponding style statistics(bottom) with respect to category labels. res1-4 denote the same as those in Figure 1. We observe that res1-4 encodes category information step by step, and the feature extractor projects images to a linearly separable feature space after the fourth residual block.

## 2  Scope of reproducibility

As follows from the introduction, MixStyle considers mixing mean and standard variation from different domains thus generating more styles implicitly, motivated by the findings that bottom layers of CNN captures style information. The authors shift and scale the mean and standard variation by AdaIN[4].

The main tested contributions are that

- Mixing styles of training instances results in novel domains being synthesized implicitly, and hence improves the generalizability of the trained model.

- The performance of MixStyle on PACS dataset is not sensitive to hyper-parameter $\alpha$, descending with the increasing value of $\alpha$.

- Applying MixStyle to multiple layers instead of single layer generally achieves better performance on category classification, except for the last residual block leading to a plunge.

## 3  Methodology

For the purpose of reproducing MixStyle, several steps were taken to cover the scope of the original paper with limited resources. Firstly, we went through the paper, grasped the meaning of method and ran with the official library Dassl.pytorch. However, the original library has implemented a great amuont of DG algorithms whose codes are redundant for us. So we extracted the core code and reorganized them into a simple demo referring to Dassl.pytorch. Secondly, in order to verify the conclusion of the original paper, we have designed extra experiments besides the original experiments carefully and reached other conclusions contradicting the original results. Thirdly, we conducted experiments on a single K80 on Colab or V100 on Huawei Cloud.

### 3.1  Model descriptions

#### 3.1.1  Background

**Instance normalization.** Instance normalization(IN) has been found effective for removing image style in style transfer models and gain a great success in AdaIN[2017]. Let $x \in \mathbb{R}^{B \times C \times H \times W}$ denoting the batch, channel, height and width of an image, respectively. IN is formulated as

$$IN(x) = \gamma \frac{x - \mu(x)}{\sigma(x)} + \beta \tag{1}$$

where $\gamma, \beta \in \mathbb{R}^C$ are learnable parameters, and $\mu(x), \sigma(x)$ are mean and standard deviation computed across the spatial dimension within each channel of each instance, i.e.

$$\mu(x)_{b,c} = \frac{1}{HW} \sum_{h=1}^{H} \sum_{w=1}^{W} x_{b,c,h,w} \tag{2}$$

and

$$\sigma(x)_{b,c} = \sqrt{\frac{1}{HW} \sum_{h=1}^{H} \sum_{w=1}^{W} (x_{b,c,h,w} - \mu(x)_{b,c})^2} \tag{3}$$

Adaptive instance normalization(AdaIN) replace the scale and shift parameters in Eq.(1) with the feature statistics of style input $y$ to achieve arbitrary style transfer:

$$AdaIN(x) = \sigma(y) \frac{x - \mu(x)}{\sigma(x)} + \mu(y) \tag{4}$$

### 3.1.2 MixStyle

MixStyle draws inspiration from AdaIN, which is designed for the purpose of regularizing CNN training by pertubing the style information of source domain training instances. To be more specific, it is plugged into CNN layers such as inserting after a residual block module in ResNet-18 or ResNet-50 as our backbone during our reproducibility.

In practice, Mixtyle is applied by integrating feature statistics of two instances with a random convex weight to simulate new styles. Given an input batch $x$, we firstly compute the standard-normalized $\widetilde{x}$. When domain labels are given, $\widetilde{x}$ is sampled uniformly from two different domains $i$ and $j$, e.g., namely $x = [x^i, x^j]$. Then we generate a new batch $x$ by a shuffling operation along both the domain dimension and in-domain instances, i.e. $\overline{x} = [Shuffle(x^j), Shuffle(x^i)]$. In cases where domain labels are unknown, x is randomly sampled from the training data, and $x = [Shuffle(x)]$. After shuffling, MixStyle computes the mixed feature statistics by computing

$$\gamma_{mix} = \lambda \sigma(x) + (1 - \lambda)\sigma(\overline{x}) \tag{5}$$

$$\beta_{mix} = \lambda \mu(x) + (1 - \lambda)\mu(\overline{x}) \tag{6}$$

In practice, $\lambda \in \mathbb{R}^B$ are instance-wise weights sampled from the beta distribution, $\lambda \sim Beta(\alpha, \alpha)$ with $\alpha \in (0, \infty)$ being a hyper-parameter. In our experiments, we investigate how the value of $\alpha$ affect the results. Finally, the mixed feature statistics are applied the style-normalized $x$ and sent to the following layers as input.

$$MixStyle(x) = \gamma_{mix} \frac{x - \mu(x)}{\sigma(x)} + \beta_{mix} \tag{7}$$

In our work, we research the effect of MixStyle on the classification task.As a popular choice, ImageNet pre-trained ResNet-18 is our main backbone and ImageNet pre-trained ResNet-50 is set as extended backbone whose results are not shown in the original paper.

### 3.2 Datasets

We reproduce the experiments on PACS dataset, which is a commonly used benchmark in domain generalization due to its larger domain shifts over VLCS dataset. PACS includes 4 domains(Photo, Sketch, Cartoon, Art), and 7 common categories 'dog', 'elephant', 'giraffe', 'guitar', 'horse', 'house', 'person'. The total number of images is 9991, but after ignoring one error image('sketch/dog/n02103406_4068-1.png') there are 9990 images in fact. For instances in domains, there are 2048 images in Art, 2344 images in Cartoon, 1670 images in Photo, 3928 images in Sketch, respectively.

| Model | Photo | Art | Cartoon | Sketch | Avg. |
|---|---|---|---|---|---|
| ResNet-18 | 95.03/95.75 | 76.76/76.61 | 76.83/75.21 | 67.39/67.39 | 79.00/78.74 |
| +MixStyle(res1234) | 92.40/92.87 | 77.20/75.20 | 75.30/69.84 | 57.03/63.87 | 75.48/75.45 |
| +MixStyle(res3) | 95.33/95.87 | 81.98/80.08 | 75.60/81.48 | 63.65/67.69 | 79.14/81.28 |
| +MixStyle(res1) | 96.11/96.11 | 79.39/77.88 | 76.02/77.86 | 66.57/71.31 | 79.52/80.79 |
| +MixStyle(res13) | 96.17/96.11 | 81.69/81.69 | 80.63/79.01 | 62.30/68.69 | 80.20/81.38 |
| +MixStyle(res23) | 95.87/95.45 | 84.52/83.20 | 77.94/81.61 | 64.15/62.98 | 80.62/80.81 |
| +MixStyle(res123) | 94.67/95.87 | 84.03/81.45 | 74.83/77.86 | 73.78/76.96 | 81.83/83.04 |
| +MixStyle(res12) | 95.69/95.69 | 84.52/83.01 | 77.01/75.90 | 70.93/71.44 | 82.04/81.51 |
| +MixStyle(res2) | 95.75/95.75 | 84.13/81.49 | 78.84/78.92 | 77.04/60.92 | 83.94/79.27 |
| +MixStyle(original results with res123) | 96.1± 0.3 | 84.1± 0.4 | 78.8± 0.4 | 75.9± 0.9 | 83.7 |

Table 1: Leave-one-domain-out generalization ascending results on PACS with the form of last-step/best-validation evaluation strategy. Feature statistics are shuffled by domain labels and the value of alpha is set to 0.1 by default.

| $\alpha$ | Photo | Art | Cartoon | Sketch | Avg. |
|---|---|---|---|---|---|
| 0.1 | 94.67/95.87 | 84.03/81.45 | 74.83/77.86 | 73.78/76.96 | 81.83/83.04 |
| 0.2 | 94.13/94.97 | 81.98/83.06 | 78.58/79.74 | 77.11/65.33 | 82.95/80.78 |
| 0.3 | 93.95/95.57 | 83.25/81.79 | 75.17/77.90 | 76.35/69.42 | 82.18/81.17 |
| 0.4 | 94.37/95.33 | 83.45/81.98 | 79.31/79.01 | 78.31/78.31 | 83.86/83.66 |
| 0.5 | 93.83/94.67 | 83.01/83.64 | 79.10/76.88 | 77.67/69.96 | 83.40/81.29 |
| 0.6 | 93.95/95.33 | 82.18/82.57 | 73.81/78.11 | 72.66/61.91 | 80.65/79.48 |

Table 2: Leave-one-domain-out generalization ascending results on PACS with the form of last-step/best-validation evaluation strategy. Feature statistics are shuffled by domain labels. the MixStyle is set after res123 of ResNet-18.

## 3.3 hyper-parameters

In the reproducibility experiments, we mainly focus on the positions $p$ after which block MixStyle layers are plugged into, and the value of $\alpha$ which controls how the beta distribution influence the value of $\lambda$. We experiment with all elements $p$ in the power set of $\{1, 2, 3\}$. And we select $alpha$ from $\{0.1, 0.2, 0.3, 0.4, 0.5, 0.6\}$ manually. The best combination of hyper-parameters are $p = \{1, 2, 3\}$ and $\alpha = 0.4$. Besides, it is worth noticing that when choosing the last step model to evaluate the results, $p = \{2\}$ and $\alpha = 0.1$ reach the peak of 83.94%. The number of total experiments is $32 \times 4$. Experimental meta-results are shown in table4.

## 3.4 Experimental setup and computational requirements

We start the experiment by re-organizing the original code from Dassl.pytorch and make it to easily modify our experiments on PACS datasets. For data preparation, we split the images from each training domains to 9(train):1(val)

| Method | Photo | Art | Cartoon | Sketch | Avg. |
|---|---|---|---|---|---|
| | | | *ResNet-18* | | |
| Baseline | 95.03/95.75 | 76.76/76.61 | 76.83/75.21 | 67.39/67.39 | 79.00/78.74 |
| CrossDomain | 94.67/95.87 | 84.03/81.45 | 74.83/77.86 | 73.78/76.96 | 81.83/83.04 |
| RandomShuffle | 95.21/95.81 | 83.30/81.93 | 76.66/77.05 | 72.30/69.20 | 81.87/81.00 |
| MixAll | 95.93/95.75 | 83.15/82.03 | 79.69/80.38 | 71.89/72.33 | 82.67/82.62 |
| | | | *ResNet-50* | | |
| Baseline | 96.89/97.43 | 86.23/86.23 | 80.93/80.93 | 70.95/73.42 | 83.78/84.50 |
| CrossDomain | 96.89/96.35 | 87.35/87.89 | 81.70/76.75 | 76.17/77.24 | 85.53/84.56 |
| RandomShuffle | 96.59/97.54 | 88.13/84.57 | 77.22/80.03 | 77.11/74.90 | 84.76/84.26 |
| MixAll | 97.07/97.31 | 86.91/86.62 | 80.29/80.29 | 75.08/75.48 | 84.84/84.93 |

Table 3: Leave-one-domain-out generalization results on PACS with the form of last-step/best-validation evaluation strategy. Feature statistics are shuffled by domain labels. the MixStyle is set after res123 of ResNet-18. And $\alpha$ is set 0.1.

| Backbone | Photo | Art | Cartoon | Sketch | Avg. |
|---|---|---|---|---|---|
| ResNet-18 | 30:23/14:35 | 29:46/12:28 | 26:27/14:03 | 26:57/12:37 | 28:23/13:25 |
| ResNet-50 | 14:59 | 14:11 | 12:59 | 12:42 | 13:42 |

Table 4: Rounded training time per experiment, measured on K80 on Colab or V100 on Huawei Cloud, with the form of minute:second.On ResNet-18, we experiment on both Colab and Huawei Cloud, thus present the form of on K80/V100. On ResNet-50, we could only experiment on Huawei Cloud, for the memory of Colab discouraging batch size of 128. The choice of hyper-parameters and Mixstyle don't affect the training and inference time.

and test on the whole held-out domain. For example, when experimenting on Photo, Art, Cartoon to the unseen Sketch, we train the training set of Photo, Art, Cartoon and validate the quality of model on validation set. After training, we choose a best model and test on the while held-out Sketch set. The final accuracy on Sketch is recorded. When each domain is held out and recorded its accuracy, the average accuracy across four domains is reported as the results to assess the effect of MixStyle compared to baselines.

For our implementation, we use the ImageNet pre-trained ResNet-18 CNN as our main backbone with one layer exporting to the evidence of each category, and ResNet-50 is used for extra experiments. It is worth noting that batch normalization is used while the paper mentioned not to use batch normalization or dropout.Using adam as optimizer, Our initial learning rate is 1e-4 and batch size is 64 for each training domain. To save computational resources, the number of epoch is 30 instead of 150 in the original library suggested. We also take cosine annealing learning rate to search a better optimal, setting the T_max parameter to 30. For training sets' transformations, we use 'random_flip', 'random_translation', 'normalize to mean of zero and standard of one' and resize to $224 \times 224$, while for test sets, we only resize them to $224 \times 224$ and normalize to mean of zero and standard of one. We use accuracy to measure the performance of the method, and all results omit the percent sign.

As is pointed out in [2], model selection plays an essential part in domain generalization. Thus, we evaluate on accordance with last step and best validation strategies simultaneously. The last step strategy chooses the model after the whole training process regardless of accuracy on validation set, and the best validation strategy considers accuracy on validation set during training process.

Experimental meta-results are shown in table 4. The original code can be found on the Github repository[1], our easy-to-modify code can be found on the Github repository[2]

# 4 Results

With respect to the insight assumption, we visualize the feature maps and the feature statistics after each residual blocks in accordance with the author's observation. However, the output of the third residual block begins to encode category information. After shuffling the output of the fourth residual block, the performance drop immediately. We reproduce the paper in totally new combination of hyper-parameters which saves 80% of the time compared with original paper and our results nearly approaches the original results with fewer epochs than that of original paper. In contrast to the original results, accuracy is sensitive to the hyper-parameter $\alpha$ as well as the places where to insert MixStyle. Besides, evaluation strategies including last-step and best-validation influence the results greatly. Lastly, when we change our backbone to ResNet-50, MixStyle still works but not that efficient as other methods.Furthermore, we test the effect of MixAll and get a pretty good results in case the hyper-parameters align with the suggested settings and a relatively stable results in our settings.

## 4.1 Results reproducing original paper

In figure 1, figure 2 and table 1 , results shows that res1-3 mainly encodes domain information while res1-4 encodes category information block by block, and it is a valid strategy to insert MixStyle after any residual block in res1-3. Restricted to computational resources, we couldn't experiment with the suggested hyper-parameters in Dassl.pytorchs. In our common hyper-parameters, table 2 indicates that the results are sensitive to the value of $\alpha$ and seemingly irregular,

---

[1]https://github.com/KaiyangZhou/Dassl.pytorch
[2]https://github.com/xuboshen/Reproducibility-challenge-2021/tree/master

which contradicts claim 2. It is supported by table 3 that cross domain performs better than random shuffling in deed. Furthermore, table 4 presents the potential in saving costs.

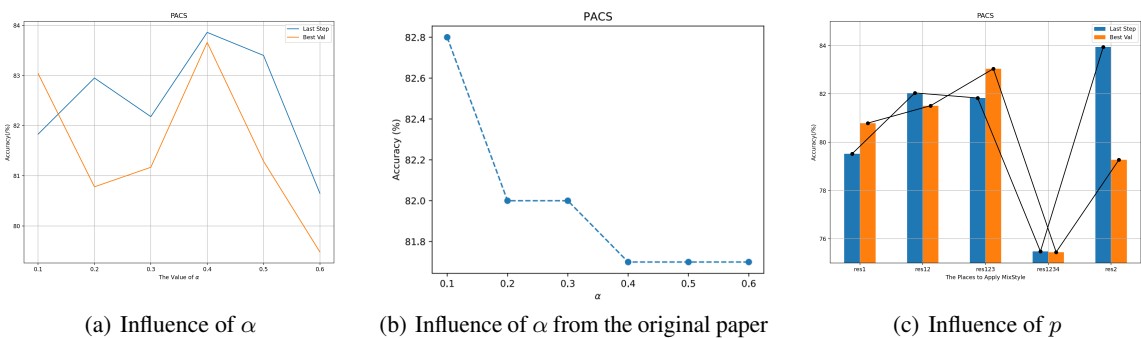

(a) Influence of $\alpha$          (b) Influence of $\alpha$ from the original paper          (c) Influence of $p$

Figure 3: Evaluation on the hyper-parameter $\alpha$ and the place $p$ to apply MixStyle on PACS

## 4.2 Results beyond original paper

Since the original paper does not specify the criterion of model selection, two mainstream model selection methods including the last-step and best-validation are exerted to test the method. In addition to experiments, we have tried all combinations of the place to insert MixStyle and found the best results are given by only insert MixStyle after res2 when choosing the last-step model as the best model, which contradicts claim 3. Moreover, because we only value MixStyle on classification task, the universal adaptability of MixStyle isn't proved, which is one of the essential argument of the original paper. Thus we investigate more on classification task by changing backbone from ResNet-18 to ResNet-50. The results in table 3 shows that the improvements are not obvious as well as that MixAll provides stability than methods of MixStyle.

## 5 Discussion

In our work, we have done as much as we could to exploit the potentials of MixStyle. To sum up, we build the code project extracted from the library Dassl.pytorch successfully and our experimental results support the main claims of the paper except for some auxiliary arguments which may be affected by the selection of hyper-parameters. And we proposed MixAll to increase the stability of MixStyle in order to make MixStyle robust to methods of model selection.

The results of GPU time indicate that MixStyle works as an easy-to-plug-in module that can easily improve generalizability of CNN models especially ResNet with little time cost. And it is indicated that the places to apply MixStyle don't matter as long as we don't insert after the fourth residual block. According to figure 3, the results are sensitive to the selection of $\alpha$. Though $\alpha$ influence results, the accuracy deviate too little to omit. We suggest choosing from $\alpha = \{0.1, 0.2, 0.3, 0.4\}$ , tuning them and getting the best combination during training stage.

Furthermore, we observed that different model selection strategies influence the results a lot. Since the more style information are mixed, the better generalizability model is, we have tried to mix all feature statistics. Results in table 3 tell us MixAll is more stable faced to different model selection strategies. By the way, we have tried to change backbone to ResNet-50 to verify the effect of MixStyle. Compared to state-of-the-art method's accuracy of 90.15%[6] and baseline of 83.78%, MixStyle only reaches 85.53% at most, showing both its little improvement and limitation.

Although MixStyle shows efficiency across different tasks, we are not knowledgeable enough to implement the remaining two task, unable to verify the tasks that MixStyle can apply to. Besides, the original paper presents a relatively small improvement on Digits-DG and Office-Home datasets. We are limited to time and computational resources provided by Huawei Cloud and daily limitations of Colab. The MixStyle still remains to be verified and improved to better utilize the domain information encoded by residual block together with the mixed category information gradually encoded.

## 5.1 What was easy

The role of MixStyle playing in ResNet is easy to understand and the way to manipulating data statistics is trivial according to AdaIN[4]. The paper is well-written and the code is easy to run, so it was simple to verify the majority of original claims. The original code provide prepared data and data pre-processing module, which saves us a great amount of time.

## 5.2 What was difficult

At first, it is confusing for us to understand why MixStyle work in domain generalization while the effect of mixup is limited. After reading related papers, including image translations and domain generalization, and visualizing the feature maps and feature statistics by t-SNE, we comprehend the insight of MixStyle and started to design experiments. Next, we spend a little more time to evaluate and design the experiments, which is of vital importance for us due to the lack of computational resources. And going through the Dassl.pytorch library has also taken us much time so that we could delete those irrelevant codes, extract the core code and make a demo to edit and experiment quickly.

## 5.3 Communication with original authors

Since we track almost all papers in DG of the authors, we are familiar with the experiment settings and Dassl.pytorch. No communication with original authors is needed. And we solve problems quickly.

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
