# OpenReview forum: "[Re]Domain Generalization with MixStyle"
_ML_Reproducibility_Challenge/2021/Fall — Reject_

### Official Review · Reviewer_wXHP · 2022-02-28

**Rating:** 4
**Confidence:** 4

**Review:**

This submission replicates the results of [MixStyle](https://arxiv.org/abs/2104.02008) for domain generalization on the PACS dataset. It confirms that MixStyle improves domain generalization performance and investigates the sensitivity of MixStyle to the mixing hyperparameter $\alpha$ as well as the position in the network that MixStyle is applied.

Strengths:
- The scope of reproducibility is clearly stated and the claims tested are clearly enumerated.
- The submission convincingly replicates the original MixStyle results. This reproduction is achieved with the original authors’ code.
- The experimental setup, including both the hyperparameters being swept and the fixed hyperparameters, is unambiguously described.
- Beyond simply experimenting with the location in the network at which MixStyle is applied and the alpha parameter as in the original paper, the submission also varies the strategy used to select the model to evaluate, either taking the last model or doing early stopping based on a validation set.

Weaknesses:
- While it is generally possible to understand the writing, it is in need of thorough proofreading and editing.
- One of the claimed contributions is evidently a new variant of MixStyle called MixAll.  However, there is no description of what MixAll consists of and how it differs from RandomShuffle.
- The test sets are very small and many of the accuracy numbers are very close to each other. Pairwise statistical tests and/or some measure of run-to-run variability would be useful to interpret which differences are likely to be real and which might not be replicable.
- The description of AdaIN and MixStyle, including both the equations and the text surrounding them, is extremely similar to the description in the original MixStyle paper. Ideally the authors would have written their own description of the method.
- It is not clear to me whether the validation set used to select the “best-validation” model consists of data from the training domains or the target domain. Assuming access to the target domain may not be fair even if it is only used for validation.

Minor:
- “Residual block” is used confusingly both here and in the original MixStyle paper. In most literature of which I am aware, a “residual block” refers to a block consisting of 2 or 3 convolutional layers that is followed by a residual connection.

---

### Official Review · Reviewer_upv9 · 2022-03-08
**Strong content, report could be improved**

**Rating:** 4
**Confidence:** 4

**Review:**

This report focused on the “Domain generalization with MixStyle” article.
The goal of the report was primarily to evaluate the reproducibility of the results presented in MixStyle when changing slightly the model and the training procedure to accommodate less powered machines. Secondly, the report proposed several extensions of the original paper. It evaluated the extent of the improvements brought by MixStyle when a different architecture is used, when the MixStyle module is applied after different layers, and when all the style statistics are used instead of just 2.

+Positive points
There are several positive points about this reproducibility report (RR).
The reproducibility authors (RA) respects the formatting of the reproducibility challenge. The reproducibility summary was rather good, and described what was attempted in his report fairly well. This RR is well built, in a logical fashion. The figures are used with parsimony, which makes reading the report relatively easy. The introduction is particularly efficient at putting in context the paper from the original authors (OA).
The RA make the sensible choice to restrict the report to classification, which is justified by the amount of computation needed to reproduce all the experiments.
The extra experiments all make a lot of sense. They are logical in the context of the paper and answer interesting research questions. This shows that the RA understood the paper they were working with and chose interesting experiments to evaluate the reproducibility of the paper, which is commendable.
Finally, the report includes valuable details such as the training time of their method and the computers used for training.

-Negative points
The reproducibility scope is not adequate.
Several changes were made to the original algorithm that are not justified properly. For instance, why add batch normalization, which could mess with the statistics that are the base of MixStyle, and especially given that the original paper seem to specifically not use BN? Another example is that the original paper mentions applying MixStyle with probability 0.5. Yet, this is not mentioned in the RR. Did the RA experiment with this hyper parameter?
Overall, it is OK to change some elements of the pipeline, but this should be mentioned clearly in the reproducibility scope so that the reader is aware of what exactly is being compared with the original paper. As it stands, the current reproducibility scope section does not describe adequately what was perform in the RR.
Moreover, the claims that are presented in the Sec. 2 could be more precise. It is for instance very difficult to test “improving the generalizability” in claim 1. Specifically, claim 3 seems closer to a post-hoc analysis rather than an actual falsifiable claim.

I found the comparison methodology imprecise at times and difficult to understand.
It is not always easy to understand what was done by the RA or by the OA, which impedes the ability to judge the reproducibility (e.g.: L.129-L.137). The report should be crystal clear on what the RA did (experiments, design choices, conclusions) and what the OA did.
I had difficulties with understanding the validation strategy and comparing it with the original paper. The authors mention doing a hyper parameter search in Sec. 3.3, but I did not understand what was the point of it since they mention using another value for alpha (0.1, the one in the original paper) than the one they found in the search. This part was confusing, potentially because a clear conclusion on this ablation was absent.
The method MixAll, introduced in this RR, could also have been described with more care; the current description is not sufficient to understand well what exactly are the modifications to the original algorithm.
In harvesting the results, the RR lacked a systematic, grounded comparison with the results for the original paper. The RA do comment subjectively on their results, but it would have been very helpful to get a comparison at a metric level (for instance, the accuracy score of experiment X was attained within X%). Moreover, MixAll is introduced by the RA to improve the “stability” but this term is never defined not measured. In general, the comments on the results are not clearly understandable from the data provided in the tables. This creates a disconnect between the actual results in the tables and the conclusions in the report.
Without precisely worded conclusions, it is challenging to get a definitive answer on the level of reproducibility of the paper.


=Conclusion and score
I think the report made a good job to set an achievable and reasonable target by restricting itself to the classification study.
The RA tried to reproduce the experiments of the original paper and proposed interesting and valuable extra experiments that bring additional knowledge to the problem.
However, I found the report a bit lacking and ultimately unfinished. Several important details are missing in the design choice, the methodology, and the conclusions of the experiments performed. Crucially, the report is lacking a well-supported conclusion on the state of reproducibility of the OP.
I recommend for now not to accept this submission as is, although I would be willing to reconsider my score if the points above were addressed.

4: Ok but not good enough - rejection


Smaller points and details

* The code was reused from the original implementation by the authors, but modified, though I was not convinced by the reasoning behind the modifications. Changing the code also changes the scope of the reproducibility and potentially introduces non-obvious differences in the code that are difficult to account for in the RR. It would have been interesting to comment on this.
* dassl.pytorch could be better introduced, the RA mention it several times but don’t precise what is the point of the library (a one liner would be enough).
* How was the range for \alpha chosen? Justifying this choice could have been useful.
* It is not clear if the experiments were computed over several seeds or only over one seed, given that this information is absent from the report and that the metrics in the tables do not have any confidence interval.
* The labelling in the tables could have been more precise: for instance, in table 3, it is not immediately obvious that Baseline is the Mixstyle version of the paper.
* The “what was difficult” section felt slightly colloquial. The “what was difficult” section voices the limits in a personal way (“[…] complicated for us students […]” rather than a general way, which was not the purpose of this section.
* At L.96, the RR defines a “x tilde” notation, but never uses it again. This notation is used in the original paper.
* The figures labels (Fig. 3 legend and title for instance) are extremely small and pretty difficult to read.

---

### Official Review · Reviewer_vT5H · 2022-03-19
**Well-written and generally thorough report attempting to reproduce the original paper. Could improve on some clarity issues.**

**Rating:** 7
**Confidence:** 4

**Review:**

The authors of the report attempt to (1) confirm the effectiveness of and (2) provide extensions to the original paper titled — “Domain Generalization with MixStyle”. The original paper addresses the problem of domain generalization (training on single/multiple source domains so as to generalize at test time to an unseen target domain) by introducing an approach titled MixStyle. Motivated from the hypothesis that a “domain” label associated with an image can be characterized by its “style” characteristics, MixStyle, essentially proposes a style-transfer approach. More specifically, unlike prior style transfer techniques, MixStyle attempts to transfer style across instances from multiple domains by computing a convex combination of feature statistics which then subsequently dictate the style transfer operation. In this report, the authors attempt to reproduce the improvements in results obtained by MixStyle over prior approaches and additionally consider an extension. In terms of overall outcome, the authors claim that their implementation achieves similar accuracy but differs a lot across checkpoint selection strategies and in general, is sensitive to hyper-parameters and the location where MixStyle is being applied. The authors borrowed part of the code from an existing open-source implementation and implemented the core method themselves. Unlike the original paper, the authors only consider one task — object recognition for their experiments.

1. The report is generally well-written and easy to follow for the most part. The authors do a decent job of outlining the core details behind MixStyle as an approach in Section 3.1 which makes it easier for a reader to grasp things quickly in a first read. The provided Reproducibility Summary, although not completely reflective of the individual conclusions drawn later from the experiments (likely due to space constraint issues), generally does a decent job of outlining the endeavor undertaken by the authors. I also particularly like that the authors spend time to outline the motivation behind MixStyle based on the 2D visualizations presented in the Introduction Section.
2. Within the limited constraints that the authors are operating in, the conducted experiments generally seem thorough with ablations outlining the layers suitable for MixStyle to be applied and what values of alpha (governing the coefficient of convex combination) are best. The authors also clearly outline the set of hyper-parameters they tried across all settings and how they split the source data into training/validation splits. They further highlight where changes had to be made in the hyper-parameters due to computational constraints.
3. In terms of results, the authors observe that the obtained results are sensitive to changes in hyper-parameters -- major diffs, I’m assuming, here are training for lesser number of iterations/epochs and starting with totally different sets of hyper-parameters. Not only sensitive, they also seem irregular. While starting with different sets of hyper-parameters doesn’t seem that much of a problem (if similar results can be achieved), I wonder if some of the instability/sensitivity could be due to not training for long enough. If that is the case, then this observation contradicting claim 2 (as mentioned in the report) is not necessarily true. I think the report would benefit from stating this observation in lieu of the two major diffs outlined above.
4. Another thing that might improve the report is formally defining/introducing MixAll as early as possible in the report. It’s hard to follow things at times when MixAll is being discussed but we’re not super-clear on what that is. Even a small blurb describing the same should be fine.
5. Other than this, the authors clearly highlight which parts of the implementation were easy/hard for them and what resources they had available for the same.

---

### Meta-Review · Area_Chair_Phbx · 2022-04-09

**Recommendation:** Reject
**Confidence:** 4

**Metareview:**

Reviewers agreed this paper was on an important topic, and praised parts of the work. They also agreed that some of the writing could be made more clear, and the experiments could be more rigorous (e.g., one reviewer asked about changes made from the original algorithm like adding batchnorm, and another highlighted that results are close and some kind of variance or statistical test would benefit the work).

---

### Decision · Program_Chairs · 2022-04-09

Reject